☵ PLOS | ONE

# Prostate-specific membrane antigen in circulating tumor cells is a new poor prognostic marker for castration-resistant prostate cancer

Naoya Nagaya[1], Masayoshi Nagata[1], Yan Lu[1], Mayuko Kanayama[1], Qi Hou[1], Zen-u Hotta[1,2], Toshiyuki China[1], Kosuke Kitamura[1], Kazuhito Matsushita[1], Shuji Isotani[1], Satoru Muto[1,3], Yoshiro Sakamoto[4], Shigeo Horie[1]*

1 Department of Urology, Juntendo University Graduate School of Medicine, Tokyo, Japan, 2 Strategic Investigation on Comprehensive Cancer Network, Interfaculty Initiative in Information Studies/Graduate School of Interdisciplinary Information Studies, the University of Tokyo, Tokyo, Japan, 3 Department of Advanced Informatics of Genetic Diseases, Juntendo University Graduate School of Medicine, Tokyo, Japan, 4 Department of Urology, Juntendo University Nerima Hospital, Tokyo, Japan

* shorie@juntendo.ac.jp

**Data Availability Statement:** All relevant data are within the manuscript.

## Abstract

The aim of this study is to elucidate the clinical significance of prostate-specific membrane antigen (PSMA) expression in circulating tumor cells (CTCs) from castration-resistant prostate cancer (CRPC) patients. We analyzed a total of 203 CTC samples from 79 CRPC patients to investigate the proportion of positive mRNA expressions at different treatment phases. Among them, we elected to focus on specimens from 56 CRPC patients who progressed on therapy and were subsequently provided a new treatment (treatment-switch cohort). In this cohort, we investigated the association between PSMA expression in CTCs and treatment response. CTCs were detected in 55/79 patients and median serum PSA in CTC-positive patients was 67.0 ng/ml. In the treatment-switch cohort of 56 patients, 20 patients were positive for PSMA in CTCs. PSMA expression was inversely associated with percentage of change in prostate-specific antigen (PSA). The median PSA progression-free survival and overall survival were significantly shorter in the PSMA-positive cohort. Furthermore, PSMA expression was predictive of poorer treatment response, shorter PSA progression-free survival and overall survival. PSMA expression in circulating tumor cells may be a novel poor prognostic marker for CRPC.

## Introduction

More than 90% of patients with metastatic treatment-naive prostate cancer initially respond to androgen deprivation therapy (ADT). However, most patients eventually progress to castration-resistant prostate cancer (CRPC), which is then treated with sequential therapy [1]. Several drugs available for sequential therapy of CRPC, such as enzalutamide (ENZ), abiraterone (ABI), docetaxel (DOC), cabazitaxel (CBZ) and radium-223, have been shown to prolong the survival of CRPC patients [2]. However, approximately 30% of CRPC patients show primary

**Funding:** The authors received no specific funding for this work.

**Competing interests:** I have read the journal's policy and the authors of this manuscript have the following competing interests: Masayoshi Nagata received research grants from Astellas and Sanofi (2017–2020). Shigeo Horie received research grants from Astellas, Sanofi (2017–2020). Shigeo Horie received as honorarium from AstraZeneca, Takeda, Janssen (2016–2019). This does not alter our adherence to PLOS ONE policies on sharing data and materials.

resistance to novel androgen receptor (AR)-targeted agents such as enzalutamide and abiraterone [3, 4]. Since tissue biopsy is invasive and cannot be performed for all tumors, liquid biopsy is drawing attention as a new testing option to overcome these shortcomings. Liquid biopsies are the blood-based analyses of non-solid biological tissues such as exosomes, cell-free DNA and circulating tumor cells (CTCs). CTCs are characterized as cancer cells that intravasate into the circulatory system from primary locations.[5]. In CRPC patients, androgen receptor splice variant 7 (AR-V7) in CTCs has the possibility to be a biomarker to predict the development of drug resistance to enzalutamide and abiraterone [6, 7]. In contrast, taxane-based chemotherapies are effective against CRPC irrespective of AR-V7 status [8, 9]. In lung cancer, genetic information of CTCs facilitates the delivery of personalized medicine [10, 11]. To optimize treatment selections and avoid unwanted adverse events, genetic analysis of CTCs would provide useful information.

Prostate-specific membrane antigen (PSMA) is a type II transmembrane glycoprotein with several enzymatic functions. Although the mechanism by which PSMA stimulates proliferation of prostate cancer remains to be elucidated [12], its expression in prostate cancer tissue correlates with cancer aggressiveness [13, 14]. Recently, the utility of PSMA PET-CT as a diagnostic tool came under the spotlight [15]. Also, PSMA is drawing attention as a target for radionuclide therapy and immunotherapy of CRPC [16–19]. Here, we investigated the PSMA expression in CTCs from CRPC patients and explored its potential as a biomarker of treatment response for CRPC.

## Materials and methods

### Patients and treatment

A total of 203 CTC samples were taken from 79 CRPC patients (1–7 samples per patient) treated at Juntendo hospital from June 2016 to June 2018. Among them, 56 CRPC patients had their CTC samples collected at the time of treatment failure, and thereafter were subjected to a new line of treatment. These patients were classified under the treatment-switch (TS) cohort. In the TS cohort, we examined whether PSMA expression in CTCs was associated with treatment response (S1 Fig). CRPC patients were treated with following agents (dosage): enzalutamide (120–160 mg/day), abiraterone (1000 mg/day), docetaxel (70 mg/$m^2$ every 3–4 weeks), cabazitaxel (20–25 mg/$m^2$ every 3–4 weeks), etoposide/ cisplatin (EP) regimen (80 mg/$m^2$ cisplatin was given on Day 1 together with 100 mg/$m^2$ etoposide on Days 1–3).

### Study design

This study was an observational study without interventions. CTC analyses were performed at the time of disease progression after primary anti-androgen therapy or subsequent therapies, and at multiple subsequent points for each patient. All clinicians were blinded to the CTC status other than AR-V7 when determining treatments. CTC-positive samples were classified according to the number of treatment lines when each sample was taken (e.g. if sample $x$ was obtained from a patient undergoing 1st-line treatments for CRPC, the sample is assigned to 1st-line category). To collectively compare mRNA expressions at the different treatment phases, treatment lines were divided into the following two groups: pre 1st-line/1st-line or 2nd-line/more. Moreover, the association between CTC profile and clinical outcomes in TS cohort were evaluated.

### Clinical outcomes in TS cohort

Prostate specific antigen (PSA) was measured in clinical routine at the clinician's discretion. The best PSA response, the percentage of change in PSA, PSA progression-free survival

(PSA-PFS) and overall survival were used as clinical outcomes in the TS cohort. Best PSA response was defined as the maximum decline in PSA that occurs at any point after treatment change, as recommended in the Prostate Cancer Clinical Trials Working Group 2 guidelines. For cases in which PSA kept rising after treatment change, the percentage of change in PSA taken for the first time after treatment change was used for assessment. When a PSA flare-up was observed, the PSA value after the flare-up was used. PSA response was defined as $\geq 50\%$ decline in PSA level from the pre-treatment level. PSA progression was defined as $\geq 25\%$ increase with an absolute increase of 2 ng/ml or more from the nadir confirmed by a second value obtained three or more weeks later [20].

## CTC analysis

We used the AdnaTest (QIAGEN, Germany) to detect CTCs in accordance with the manufacturer's protocol [6, 7]. 5 ml of the patient's blood was drawn into EDTA-3K collection tubes, followed by RNA extraction with antibody-conjugated magnetic beads using the AdnaTest ProstateCancerSelect. Then, mRNA was extracted by the AdnaTest ProstateCancerDetect. Extracted mRNA was subjected to reverse transcription using the Sensiscript Reverse Transcriptase Kit (QIAGEN). Expressions of PSMA, AR-V7, AR, and Epidermal Growth Factor Receptor (EGFR) in CTCs were examined by reverse transcription polymerase chain reaction (RT-PCR). The AdnaTest PrimerMix ProstateDetect was used for amplification of PSA, PSMA, and EGFR (PCR condition for PSA, PSMA, and EGFR: 95°C for 15 min, 42 cycles of 94°C for 30 sec, 61°C for 30 sec, 72°C for 30 sec, followed by 10 min of extension). The AdnaTest PrimerMix AR-Detect was used for amplification of AR (PCR condition for AR: 95°C for 15 min, 35 cycles of 94°C for 30 sec, 60°C for 30 sec, 72°C for 60 sec, followed by 10 min of extension). The manufacturer defined the CTC presence as any one of PSMA, PSA, AR or EGFR expression. It was confirmed from our experiments that samples positive for any one of AR, PSMA or EGFR are 100% positive for PSA. Thus, we concluded that PSA positivity is a common denominator and defined successful CTC detection as positive PSA expression in this study. The primer set and a PCR condition for AR-V7 RT-PCR is as follows; AR-V7 primer set designed to yield 125-bp AR-V7-specific band: 5'-CCATCTTGTCGTCTTCGGA AATGTTA-3' and 5'-TTTGAATGAGGCAAGTCAGCCTTTCT-3' (PCR condition for AR-V7: 95°C for 5 min, 39 cycles of 95°C for 10 sec, 58°C for 30 sec, 72°C for 30 sec, followed by 10 min of extension). Amplified PCR products were electrophoresed and visualized by the DNA 1K Experion automated electrophoresis system (Bio-Rad, CA, USA). To evaluate gene expression, the fluorescence intensity scale was set to "scale to local" (default setting), and any visible bands under this condition with detectable peaks were considered positive.

## The Cancer Genome Atlas (TCGA) data analysis

To complement our small sample size, we utilized another independent cohort from TCGA that is open-access and provides both genomic and clinical data. The Cancer Genome Atlas Research Network showed comprehensive molecular analysis of primary prostate cancer. This cohort contained 333 prostate cancer patients, for which both overall survival and mRNA expression data (PSMA, AR, AR-V7, and EGFR) were available for 316 patients [21]. The data and analysis results are available on the cBioPortal for Cancer Genomics (https://www.cbioportal.org/). To evaluate the correlation between mRNA expression of primary prostate cancer and clinical outcomes, we divided the cohort into two groups based on the presence of AR-V7 mRNA expression: AR-V7 positive (n = 80) and AR-V7 negative (n = 236). As for the other mRNA expressions, the cohort was divided in half into the high expression (n = 158) and the low expression group (n = 158).

## Statistical analysis

Statistical analyses were performed using the Fisher's exact test for categorical variables, and the Wilcoxon Mann-Whitney for continuous variables. The PSA-PFS and overall survival analyses were done with the Kaplan-Meier plot, and differences were compared with the log-rank test. Multivariable analyses were performed using multiple regression analysis and Cox proportional hazard model. Statistical significance was defined as $P < 0.05$.

## Ethics

This study was approved by the institutional review board of Juntendo hospital (admission number: 14–052), and all experiments were carried out in accordance with approved guidelines. All participants submitted written informed consent.

## Results

### Characteristics of CTC and mRNA expressions at different treatment phases

A total of 203 samples from 79 CRPC patients were analyzed (Fig 1A). CTCs were detected in 127 samples (63%). The median age of all CRPC patients was 73 (range: 50–89). The median serum PSA at the time of CTC analysis was 18.8 ng/ml for all of 203 samples, 67.0 ng/ml for CTC-positive samples (n = 127), and 2.3 ng/ml for CTC-negative samples (n = 76). PSA was significantly higher in the CTC-positive samples ($P < 0.001$). The proportion of mRNA expressions in these CTC-positive 127 samples were as follows: 63% were positive for PSMA, 71% positive for AR, 25% positive for AR-V7, and 22% positive for EGFR. Among 127 CTC-positive samples, 58 of them were obtained from patients undergoing pre 1st-line /1st-line therapy, while 69 samples were obtained from patients receiving 2nd-line/more. The details of CRPC treatments at the point of sample collection were indicated in the charts. In the 2nd-line/more groups, 30% of samples originated from patients receiving luteinizing hormone-releasing hormone (LH-RH) monotherapy. These patients were under the best supportive care and did not receive any additional treatments for CRPC. Interestingly, PSMA, AR, AR-V7 and EGFR levels were higher in samples from patients receiving 2nd-line/more therapy ($P < 0.05$) (Fig 1B).

### Patients' characteristics of the TS cohort

Fifty-six CRPC patients were analyzed in the TS cohort (Fig 2A). Details of administrated treatments were as follows: AR-targeted therapy for 36 patients (enzalutamide for 32 patients, abiraterone for 4 patients), systemic chemotherapy for 20 patients (docetaxel for 13 patients, cabazitaxel for six patients and etoposide plus cisplatin for one patient). Thirty-nine patients (70%) were positive for CTCs. Overall patients' characteristics classified by the presence of CTC are summarized in Table 1. Baseline PSA ($P < 0.0001$), alkaline phosphatase (ALP) ($P = 0.025$) and Bone Scan Index (BSI) ($P = 0.011$) were significantly higher in the CTC-positive cohort, indicating that CTCs are more likely to be detected in advanced diseases. In addition, the prior use of abiraterone was significantly associated with the presence of CTCs ($P = 0.023$), and time since diagnosis was significantly shorter in the CTC-positive cohort ($P = 0.032$). The presence of CTCs was not significantly correlated with the percentage of change in PSA (median change: -80.5% vs -75.3%, $P = 0.233$, Fig 2B) and overall survival ($P = 0.685$, Fig 2C) after treatment switch. In contrast, the median PSA-PFS was significantly shorter for the CTC-positive cohort ($P = 0.005$, Fig 2C).

In CTC-positive patients, 20 out of 39 patients (51.3%) were positive for PSMA. Baseline characteristics of 39 CTC-positive patients classified by PSMA status are summarized in

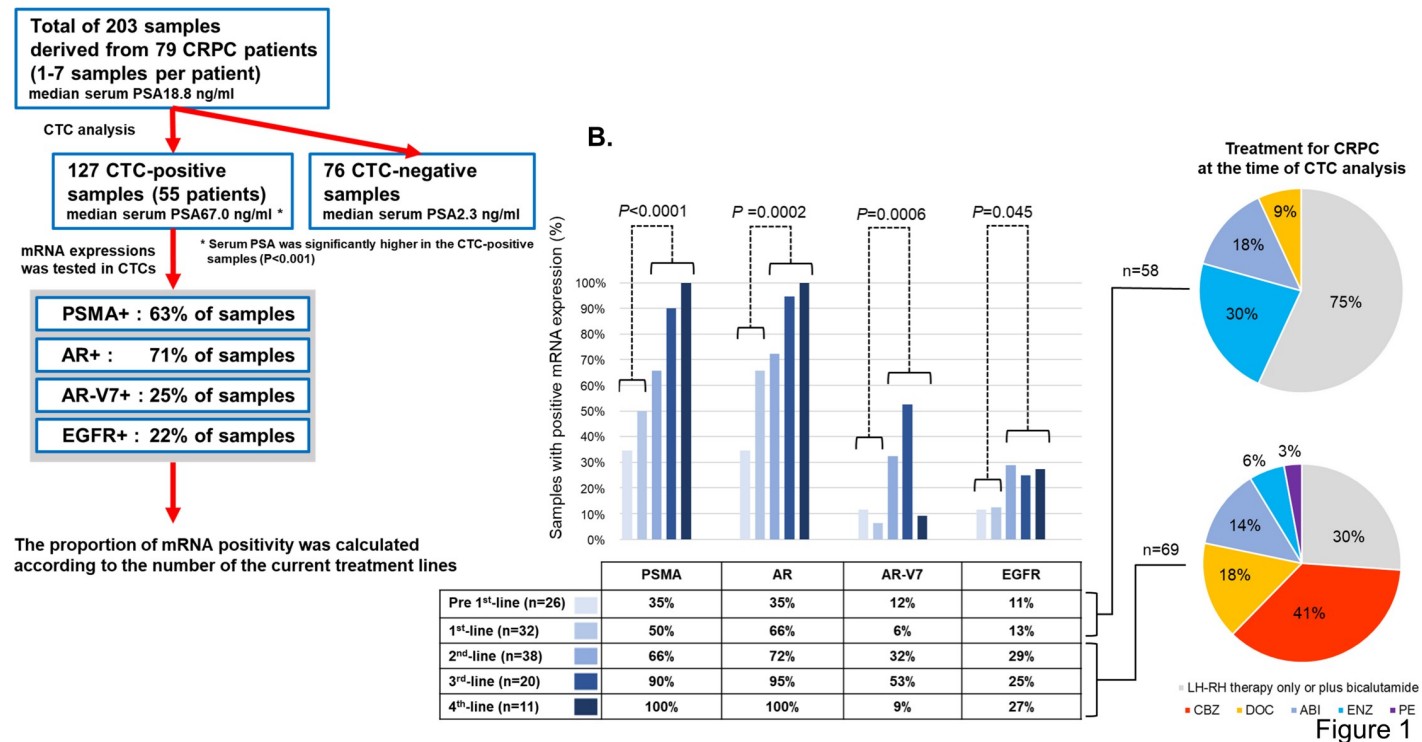

**Fig 1. Comprehensive analysis of all samples and mRNA expressions at a different treatment phase. A.** A flowchart for all samples' analysis. The number of CTC detection and mRNA positivity are indicated in the box. Out of 203 CTC samples in total, 127 of them were positive for CTCs. **B.** 127 CTC-positive samples were classified according to the number of treatment when each sample was taken. Treatment lines were divided into two groups: pre 1st-line/1st-line or 2nd-line/more. The proportions of positive mRNA expressions were compared between these two groups by the Fisher's exact test. PSMA, AR, AR-V7, and EGFR were significantly more expressed in 2nd-line/more group ($P < 0.05$).

Table 2. There were significant differences in baseline characteristics between the PSMA-positive and the negative cohorts such as age ($P = 0.045$), baseline PSA ($P = 0.023$) and the prior use of enzalutamide ($P = 0.008$).

## Treatment response according to PSMA expressions in CTCs

In 39 CTC-positive patients, PSMA expression was inversely associated with the percentage of change in PSA (median change: -90.9% vs. -13.8%, $P < 0.0016$). Whereas other gene expressions such as AR, EGFR and AR-V7 were not significantly associated with the percentage of change in PSA (Fig 3). Next, we schematically summarized PSA change, mRNA expressions in CTCs and treatment history of these 39 CTC-positive patients according to PSMA status (Fig 4). Among them, seven of the eight patients who showed PSA progression, irrespective of AR-V7 status, were positive for PSMA in CTCs. Meanwhile, PSA response was observed in 23 cases, 17 of which were negative for PSMA expression. Three of the six AR-V7-positive patients who were treated with cytotoxic systemic chemotherapy showed PSA decline, and two of them showed PSA response. As shown in the Kaplan-Meier plot, the median PSA-PFS was significantly shorter in PSMA-positive cohort (Fig 5A) (12 weeks for PSMA-positive vs. 30 weeks for negative, $P = 0.008$ by the log-rank test), and the median overall survival was also significantly shorter in PSMA-positive cohort (Fig 5A) (13 months for PSMA-positive vs. 27 months for negative, $P = 0.010$ by the log-rank test). To investigate the association between PSMA and treatment response in different treatment groups, CTC-positive patients were

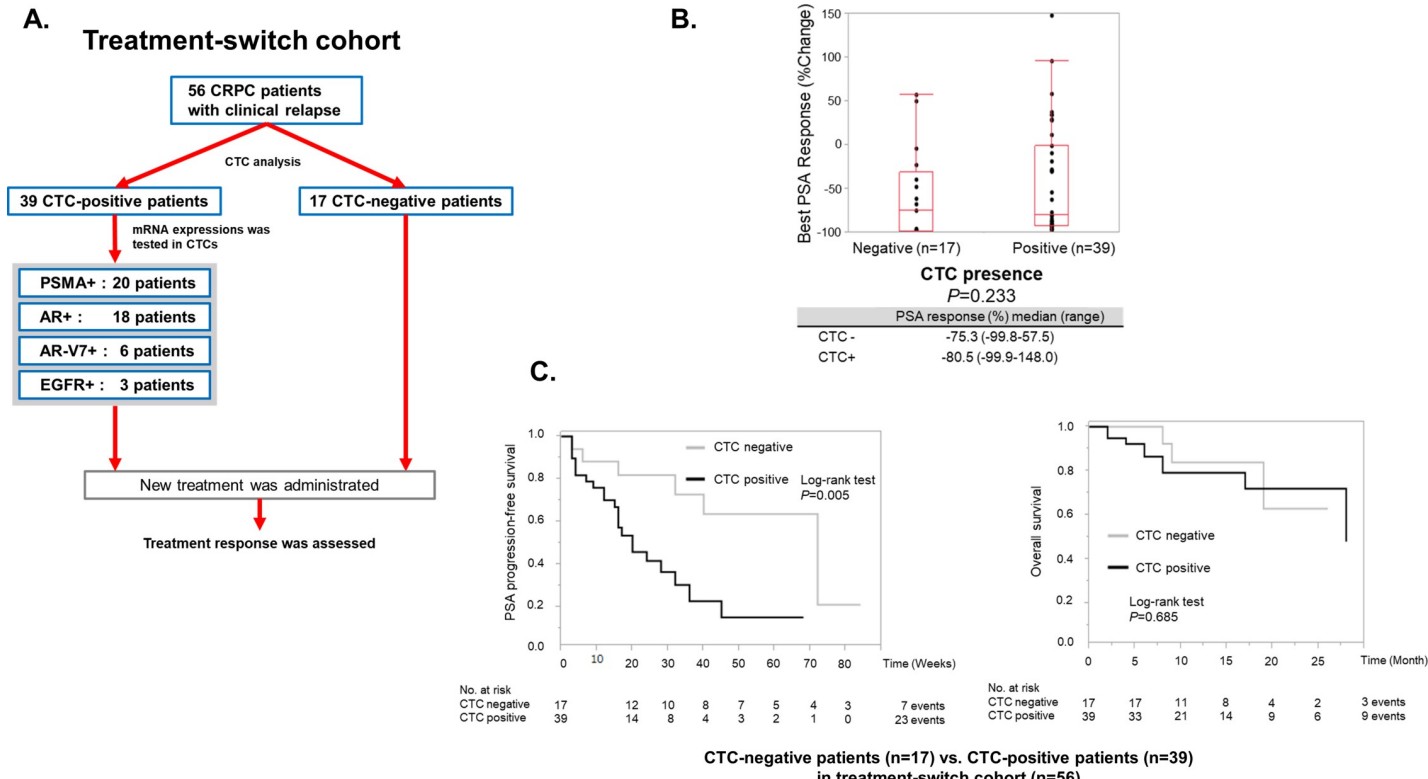

**Fig 2. Overview of a treatment-switch cohort and the association between CTC presence and treatment response. A.** A flowchart for the "Treatment-switch cohort". This cohort includes 56 patients who had their CTC samples collected at the time of recurrence and were given new treatments thereafter. The number of CTC detection and mRNA positivity are indicated in the box. **B.** The Wilcoxon Mann-Whitney test was used to assess the association between the best PSA response or the percentage of change in PSA and CTC presence. The presence of CTCs was not significantly correlated with the percentage of change in PSA (median change: -75.3% vs -80.5%, *P* = 0.233). **C.** The Kaplan-Meier plot of PSA-PFS and Overall survival was drawn based on CTC presence. The differences were compared with the log-rank test. The median PSA-PFS was 20 weeks (95% CI, 15 to 32) for the CTC-positive cohort, while it was 72 weeks (95% CI, 32 to -) for the negative cohort. PSA-PFS was significantly shorter in the CTC-positive cohort by the log-rank test (*P* = 0.005). The median Overall survival was 28 months (95% CI, 17 to -) for the CTC-positive cohort, while it was not reached (95% CI, 9 to -) for the negative cohort. Overall survival was not significantly shorter in the CTC-positive cohort by the log-rank test (*P* = 0.685).

classified as either anti-androgen therapy cohort or chemotherapy cohort. Although not statistically significant, PSA-PFS in chemotherapy cohort tended to be shorter in the PSMA-positive cohort (*P* = 0.099) (Fig 5B).

## PSMA expression in CTC as a poor prognostic biomarker

Multiple regression analysis showed that PSMA status in CTCs was an independent predictor of the best PSA response/the percentage of change in PSA after the treatment switch (*P*<0.001) (Table 3). Furthermore, the Cox proportional hazard model showed that PSMA status together with Gleason sum are useful in predicting a shorter PSA-PFS (hazard ratio with PSMA positivity = 4.02; 95% CI, 1.33 to12.8; *P* = 0.014) (Table 4), and PSMA status is also useful to predict a shorter overall survival (hazard ratio with PSMA positivity = 7.62; 95% CI, 1.08 to153; *P* = 0.040) (Table 5).

## High PSMA expression in TCGA cohort

To complement our small sample size, we further investigated the correlation between PSMA expression in primary prostate cancer tissue and overall survival in TCGA cohort, in which PSMA, AR, AR-V7 and EGFR mRNA expression data were available for 316 prostate cancer

**Table 1. Baseline characteristics of all CRPC patients in treatment-failure cohort.**

| Baseline Characteristics | All Patients (*N* = 56) | CTC-Negative Patients (*N* = 17) | CTC-Positive Patients (*N* = 39) | *P* |
|---|---|---|---|---|
| Age (years) median (range) | 73 (51–89) | 74 (60–84) | 73 (51–89) | 0.708 |
| Gleason sum at diagnosis median (6-8/9-10) *N* (%) | 6–8:39 (69.6%) / 9–10:14 (25%) | 6–8:10 (58.8%) / 9–10:6 (35.3%) | 6–8:29 (74.4%) / 9–10:8 (22.2%) | 0.311 |
| Tumor stage at diagnosis (T1-3/T4) *N* (%) | T1-3:35 (62.5%) / T4:13 (23.2%) | T1-3:12 (70.6%) / T4:2 (11.8%) | T1-3: 23 (59.0%) / T4:11 (28.2%) | 0.434 |
| Time since diagnosis (y) median (range) | 2.5 (0.50–16.0) | 3.6 (0.83–16.0) | 2.0 (0.50–15.0) | **0.032** |
| Presence of visceral meta (y/n) *N* (%) | Yes 16 (28.6%) / No 40 (71.4%) | Yes 5 (29.4%) / No 12 (70.6%) | Yes 11 (28.2%) / No 28 (71.8%) | >0.999 |
| Presence of bone meta (y/n) *N* (%) | Yes 42 (75%) / No 14 (25%) | Yes 12 (70.6%) / No 5 (29.4%) | Yes 30 (76.9%) / No 9 (23.1%) | 0.739 |
| Presence of lymph node meta (y/n) *N* (%) | Yes 24 (42.9%) / No 32 (57.1%) | Yes 6 (35.3%) / No 11 (64.7%) | Yes 18 (46.2%) / No 21 (53.8%) | 0.561 |
| Baseline BSI median (%) (range) | 0.25 (0–9.86) | 0.07 (0–3.43) | 0.80 (0–9.86) | **0.011** |
| Baseline Lactate Dehydrogenase (LDH) median (U/L) (range) | 202 (113–840) | 200 (113–250) | 208 (136–840) | 0.151 |
| Baseline ALP median (U/L) (range) | 263 (86–1114) | 192 (86–406) | 287 (112–1114) | **0.025** |
| Baseline PSA median (ng/ml) (range) | 20.4 (0.33–799) | 3.12 (0.33–48.6) | 40.4 (0.92–799) | **<0.0001** |
| Treatment lines for CRPC (Pre-1st/2nd-4th) | Pre-1st:49 (87.5%) / 2nd-4th:7 (12.5%) | Pre-1st:17 (100%) / 2nd-4th:0 (0%) | Pre-1st:32 (82.1%) / 2nd-4th:7 (17.9%) | 0.088 |
| Prior use of ENZ (y/n) *N* (%) | Yes 11 (19.6%) / No 45 (80.4%) | Yes 1 (5.9%) / No 16 (94.1%) | Yes 10 (25.6%) / No 29 (74.4%) | 0.144 |
| Prior use of ABI (y/n) *N* (%) | Yes 10 (17.9%) / No 46 (82.1%) | Yes 0 (0%) / No 17 (100%) | Yes 10 (29.6%) / No 29 (70.4%) | **0.023** |
| Prior use of DOC (y/n) *N* (%) | Yes 13 (23.2%) / No 43 (76.8%) | Yes 4 (23.5%) / No 13 (76.5%) | Yes 9 (23.1%) / No 30 (76.9%) | >0.999 |
| Prior use of CBZ (y/n) *N* (%) | Yes 0 (0%) / No 56 (100%) | Yes 0 (0%) / No 17 (100%) | Yes 0 (0%) / No 39 (100%) | - |
| Type of local treatment *N* (%) (surgery) | Yes 5 (8.9%) / No 51 (91.1%) | Yes 2 (11.8%) / No 15 (88.2%) | Yes 3 (7.7%) / No 36 (92.3%) | 0.633 |
| Type of local treatment *N* (%) (radiation) | Yes 7 (12.5%) / No 49 (87.5%) | Yes 4 (23.5%) / No 13 (76.5%) | Yes 3 (7.7%) / No 36 (92.3%) | 0.182 |

The baseline characteristics of all patients before treatment change were indicated in the "All Patients" column. They were subsequently classified based on the absence or presence of CTCs, and the same variables were used to compare the CTC-positive and negative cohorts. Fisher's exact test was used for categorical variables, and Wilcoxon Mann-Whitney test was used for continuous variables.

patients. Patients with prostate cancer expressing higher PSMA showed significantly and shorter overall survival ($P$ = 0.034) (Fig 6), whereas such tendency was not obvious for the expressions of other genes.

## Serial CTC analysis in individual patients

mRNA expressions in CTCs were serially examined along the treatment course at regular intervals. mRNA status including PSMA changed during treatment courses. Patient 26 was positive for PSMA as early as at the time of diagnosis (Fig 7A). He developed CRPC within five months after the initiation of the primary combined androgen blockade (CAB). Given the positive-AR-V7 expression and the results of earlier studies recommending the early introduction of docetaxel for rapidly progressing CRPC [22, 23], we introduced docetaxel soon after the primary CAB. After the temporary negative conversion of PSMA, AR-V7 and EGFR, docetaxel had to be suspended because of adverse events. Even though cabazitaxel was given as the second-line chemotherapy, PSMA, AR-V7 and EGFR turned out positive again with a gradual PSA increase. Eventually, chemotherapy was discontinued because of a lumber bone fracture

**Table 2. Baseline characteristics of 39 CTC-positive CRPC patients.**

| Baseline Characteristics | CTC-Positive Patients (N = 39) | PSMA Negative (N = 19) | PSMA Positive (N = 20) | P |
|---|---|---|---|---|
| Age (years) median (range) | 73 (51–89) | 71 (51–88) | 75 (54–89) | **0.045** |
| Gleason sum at diagnosis median (6-8/9-10) N (%) | 6–8:29 (74.4%) / 9–10:8 (22.2%) | 6–8:14 (73.7%) / 9–10:5 (26.3%) | 6–8:15 (75.0%) / 9–10:3 (15.0%) | 0.692 |
| Tumor stage at diagnosis (T1-3/T4) N (%) | T1-3: 23 (59.0%) / T4:11 (28.2%) | T1-3:11 (57.9%) / T4:6 (31.6%) | T1-3:12 (60.0%) / T4:5 (25.0%) | >0.999 |
| Time since diagnosis (y) median (range) | 2.0 (0.50–15.0) | 1.5 (0.50–8.6) | 4.0 (0.50–15.0) | 0.112 |
| Presence of visceral meta (y/n) N (%) | Yes 11 (28.2%) / No 28 (71.8%) | Yes 6 (31.6%) / No 13 (68.4%) | Yes 5 (25.0%) / No 15 (75.0%) | 0.731 |
| Presence of bone meta (y/n) N (%) | Yes 30 (76.9%) / No 9 (23.1%) | Yes 14 (73.7%) / No 5 (26.3%) | Yes 16 (80.0%) / No 4 (20.0%) | 0.716 |
| Presence of lymph node meta (y/n) N (%) | Yes 18 (46.2%) / No 21 (53.8%) | Yes 7 (36.8%) / No 12 (63.2%) | Yes 11 (55.0%) / No 9 (45.0%) | 0.340 |
| Baseline BSI median (%) (range) | 0.80 (0–9.86) | 0.89 (0–6.48) | 0.6 (0–9.86) | 0.364 |
| Presence of AR-V7 in CTC (y/n) N (%) | Yes 6 (15.4%) / No 33 (84.6%) | Yes 1 (5.3%) / No 18 (94.7%) | Yes 5 (25.0%) / No 15 (75.0%) | 0.181 |
| Baseline LDH median (U/L) (range) | 208 (136–840) | 221 (136–432) | 205 (151–840) | 0.910 |
| Baseline ALP median (U/L) (range) | 287 (112–1114) | 351 (112–999) | 280 (115–1114) | >0.999 |
| Baseline PSA median (ng/ml) (range) | 40.4 (0.92–799) | 23.5 (0.92–578) | 112 (4.16–799) | **0.023** |
| Treatment lines for CRPC (Pre-1st/2nd-4th) | Pre-1st:32 (82.1%) / 2nd-4th:7 (17.9%) | Pre1st:18 (94.7%) / 2nd-4th:1 (5.3%) | Pre1st:14 (70.0%) / 2nd-4th:6 (30.0%) | 0.091 |
| Prior use of ENZ (y/n) N (%) | Yes 10 (25.6%) / No 29 (74.4%) | Yes 1 (5.3%) / No 18 (94.7%) | Yes 9 (45.0%) / No 11 (55.0%) | **0.008** |
| Prior use of ABI (y/n) N (%) | Yes 10 (29.6%) / No 29 (70.4%) | Yes 3 (15.8%) / No 16 (84.2%) | Yes 7 (35.0%) / No 13 (65.0%) | 0.273 |
| Prior use of DOC (y/n) N (%) | Yes 9 (23.1%) / No 30 (76.9%) | Yes 5 (26.3%) / No 14 (73.7%) | Yes 4 (20.0%) / No 16 (80.0%) | 0.716 |
| Prior use of CBZ (y/n) N (%) | Yes 0 (0%) / No 39 (100%) | Yes 0 (0%) / No 19 (100%) | Yes 0 (%) / No 20 (100%) | - |
| Type of local treatment N (%) (surgery) | Yes 3 (7.7%) / No 36 (92.3%) | Yes 2 (10.5%) / No 17 (89.5%) | Yes 1 (5.0%) / No 19 (95.0%) | 0.605 |
| Type of local treatment N (%) (radiation) | Yes 3 (7.7%) / No 36 (92.3%) | Yes 1 (5.3%) / No 18 (94.7%) | Yes 2 (10.0%) / No 18 (90.0%) | >0.999 |

The baseline characteristics of 39 CTC-positive patients before treatment change were indicated in the "CTC-Positive Patients" column. They were subsequently classified according to PSMA status, and the same variables were used to compare the PSMA-positive and negative cohorts. Fisher's exact test was used for categorical variables, and Wilcoxon Mann-Whitney test was used for continuous variables.

and palliative care was started. On the contrary, shown in Fig 7B is a case with positive conversion of PSMA expression during treatment courses. Patient 19 who was initially negative for both PSMA and AR-V7 responded to enzalutamide. Then, this patient showed PSA progression with a concurrent positive conversion of PSMA. In spite of systemic chemotherapies, serum PSA increased gradually and PSMA remained positive.

## Discussion

In this study, CTCs were detected in 63% of samples derived from CRPC patients. The detection rate was slightly lower than that in the previous report [7]. This may be ascribed to the fact that our cohort contained patients at relatively early stages of the disease. Furthermore, we showed that CTCs are more likely to be detected in patients with high serum PSA levels like

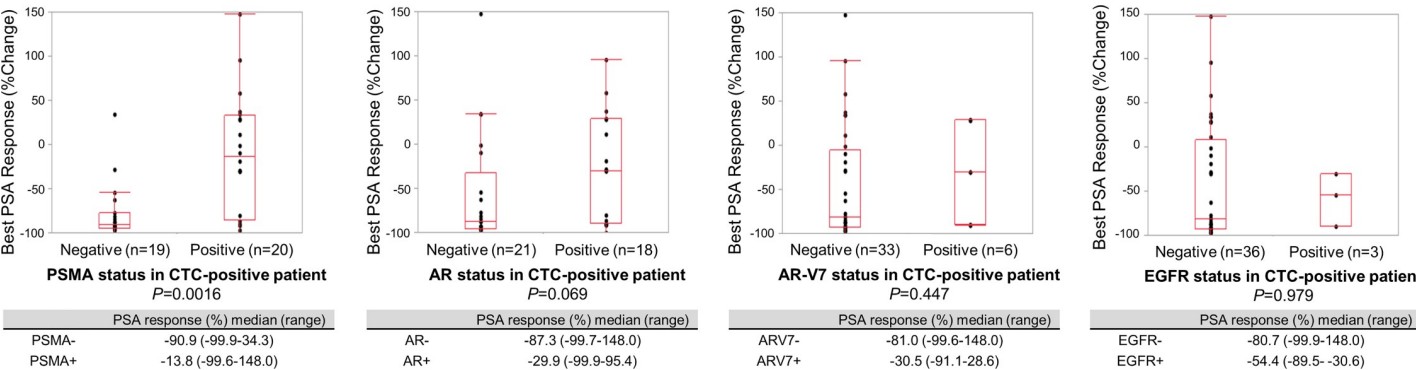

**Fig 3. The association between CTCs' gene expression and treatment response.** The Wilcoxon Mann-Whitney test was used as a univariate analysis to assess the association between the best PSA response or the percentage of change in PSA and CTCs' gene expressions (PSMA, EGFR, AR, and AR-V7). Only PSMA was inversely correlated with the percentage of change in PSA with a statistical significance (median change: -90.9% vs -13.8%, $P$ = 0.0016). AR expression approached borderline statistical significance (median change: -87.3% vs -29.9%, $P$ = 0.069).

previous report [7], and CTCs from patients treated with more than 2nd line therapy significantly express more PSMA, AR, AR-V7 and EGFR. Notably, PSMA expression was proven to predict poor clinical outcomes based on multivariate analyses. Although previous reports have shown that CTC-positive patients have significantly shorter overall survival, the presence of CTC was not a good indicator of overall survival in our study [7, 24]. Other reports using AdnaTest define the presence of CTC as the expression of any of PSA, PSMA, EGFR, or AR. Meanwhile, in our study, prostate cancer CTCs were categorized into only two types: PSA + / PSMA + or PSA + / PSMA-. This means that our CTC stratification based on PSMA is more detailed than other inclusive definition of CTC positivity. Thus, a possible explanation for this discrepancy is that CTC stratification based on PSMA better predicts overall survival than CTC positivity alone in a relatively small cohort like this study.

Given that PSA or AR-V7 expression alone cannot predict response to forthcoming treatments, PSMA expression in CTC should also be considered as one of the poor prognostic factors in CRPC.

Increased detection rates of PSMA, AR, AR-V7 and EGFR in CTC samples taken from heavily treated CRPC patients possibly indicate treatment-resistance mechanisms. Earlier studies have shown that AR aberrations in CTCs and cell-free DNA such as alternatively spliced forms, copy number gains, point mutations and structural variations increased among CRPC patients who had been previously treated with anti-androgen therapies [25]. In line with this, we confirmed increased expression of AR and AR-V7 in the latter phase of treatments. In a similar vein, PSMA expression increased in proportion to the number of treatment lines, giving credibility to the notion that PSMA is also indicative of some treatment-resistant mechanisms. To the best of our knowledge, this study is the first to evaluate the significance of PSMA expression in CTCs of CRPC patients.

PSMA was reported to promote cancer progression and malignant transformation at least in part via neovascularization [12, 13]. Immunohistochemical analysis revealed increased PSMA expression in primary and metastatic lesions of prostate cancer compared with normal prostate, and PSMA expression was reported to correlate with Gleason score and cancer aggressiveness [13, 14] [26]. As opposed to these data, in this study, there was no significant association between Gleason score and PSMA expression in CTC (Table 2), suggesting that PSMA in CTCs is an independent poor prognostic marker.

A previous study has shown that gene expressions in CTCs, such as AR-V7, change depending on the treatment status [27]. Likewise, our results showed that the PSMA is very reflective

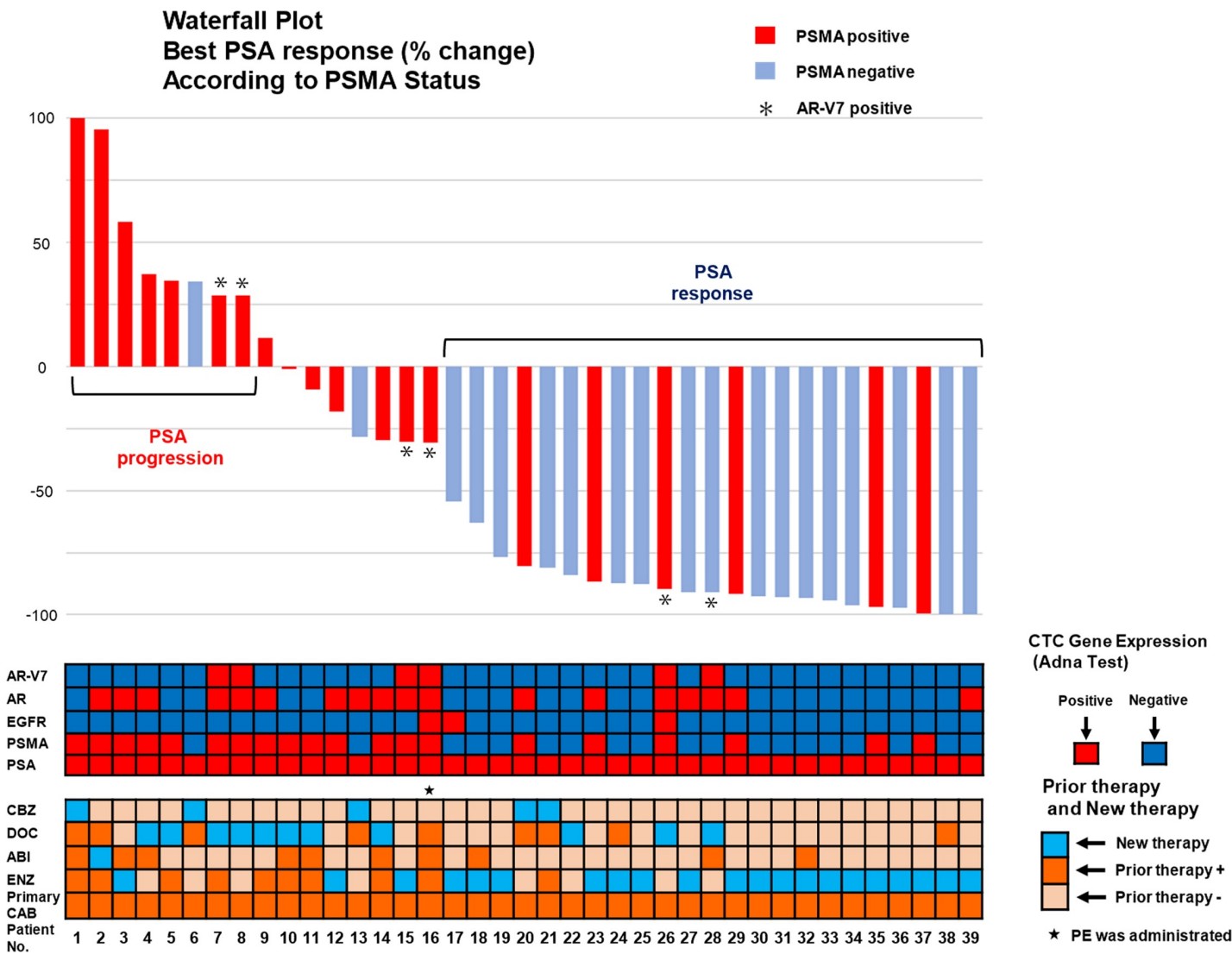

**Fig 4. The overall profiles of the Treatment-switch cohort, waterfall plot of PSA change.** The overview of 39 CTC-positive patients in the Treatment-switch cohort is shown. The overall results of CTC analysis, prior treatment history, and a waterfall plot of the best PSA response or the percentage of change in PSA were schematically summarized. The meaning of the color-coding is as indicated. AR-V7 positive cases are designated as asterisk in the waterfall plot.

of a treatment status and aggressiveness of the disease. In addition, we observed an increased PSMA expression in patients previously treated with enzalutamide (Table 2). This is possibly because AR-signaling-targeted therapies induce PSMA expression as reported by earlier *in vitro* and *in vivo* studies [28–30].

The limitation of our study, firstly, was that the timing of sample collection during CRPC treatment was not uniform. In order to deal with this limitation, we classified CTC-positive samples according to the number of treatment lines based on the assumption that treatment resistance is acquired through sequential treatments [31, 32]. The second limitation is the small sample size. Although we analyzed total of 127 CTC-positive samples, these samples were obtained from 55 patients. Likewise, TS cohort was small (n = 56). An analysis of a larger cohort is desirable to investigate the PSMA status in the respective treatment group. It is also necessary to clarify the oncological function of PSMA molecule hereafter by *in vitro* and *in vivo* studies.

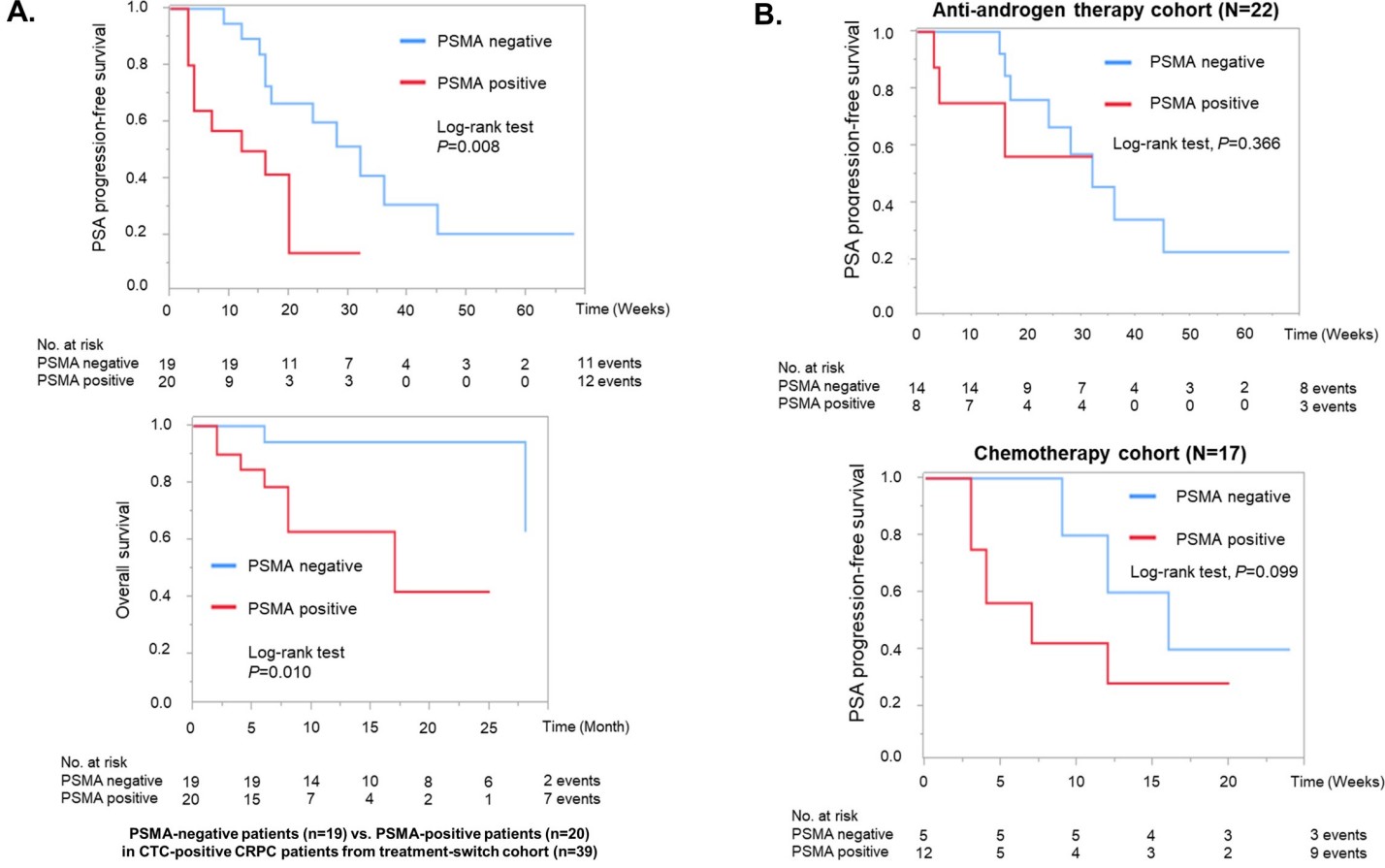

**Fig 5. PSA-PFS and overall survival according to the PSMA expression in CTCs. A.** The Kaplan-Meier plots of PSA-PFS and overall survival were drawn according to the PSMA status in CTCs. The differences were compared with the log-rank test. **B.** The Kaplan-Meier plots of PSA-PFS were plotted according to the PSMA status in anti-androgen therapy and chemotherapy cohort respectively.

In conclusion, we have shown that CTCs obtained from advanced diseases express more PSMA. Furthermore, our study is the first to provide the evidence of an association between PSMA expression in CTCs and poor treatment response in CRPC. PSMA expression independently predicts both poor treatment response and shorter PSA-PFS and overall survival, suggesting that PSMA in CTCs may be a novel prognostic biomarker.

**Table 3. Multiple regression analysis: Prediction of the best PSA response or the percentage of change in PSA from PSMA, baseline PSA, baseline BSI, and Gleason sum (39 CTC-positive patients).**

| Variable | $t$ value | $P$ value |
|---|---|---|
| Model: $R^2$ = 0.345, F = 4.090, P = 0.008 | | |
| PSMA (positive/negative) | 3.39 | **0.001** |
| Baseline PSA (ng/ml) | 0.15 | 0.883 |
| Baseline BSI (%) | -0.91 | 0.367 |
| Gleason sum (6-8/9-10) | 0.93 | 0.359 |

The result of ANOVA for this model is indicated in the upper table ($P$ = 0.008). $R^2$ is $R$ square for multiple regression equation. $F$ is the probability of $F$ associated with multiple regression.

**Table 4. Cox proportional hazard model: Prediction of PSA progression based on PSMA and baseline PSA, baseline BSI, and Gleason sum (39 CTC-positive patients).**

| Variable | Hazard ratio (95% CI) | *P* value |
|---|---|---|
| PSMA (positive/negative) | 4.02 (1.33–12.8) | **0.014** |
| Baseline PSA (ng/ml) | 1.48 (0.21–8.43) | 0.652 |
| Baseline BSI (%) | 3.71 (0.31–39.4) | 0.675 |
| Gleason sum (6-8/9-10) | 3.29 (1.06–9.89) | **0.039** |

PSMA expression in CTCs and Gleason sum were significantly predictive of PSA-PFS in comparison with baseline PSA and baseline BSI.

**Table 5. Cox proportional hazard model: Prediction of overall survival based on PSMA and baseline PSA, baseline BSI, and Gleason sum (39 CTC-positive patients).**

| Variable | Hazard ratio (95% CI) | *P* value |
|---|---|---|
| PSMA (positive/negative) | 7.62 (1.08–153) | **0.040** |
| Baseline PSA (ng/ml) | 1.41 (0.06–17.7) | 0.802 |
| Baseline BSI (%) | 1.45 (0.09–21.6) | 0.778 |
| Gleason sum (6-8/9-10) | 1.85 (0.26–10.4) | 0.507 |

PSMA expression in CTCs was significantly predictive of overall survival in comparison with baseline PSA, baseline BSI and Gleason sum.

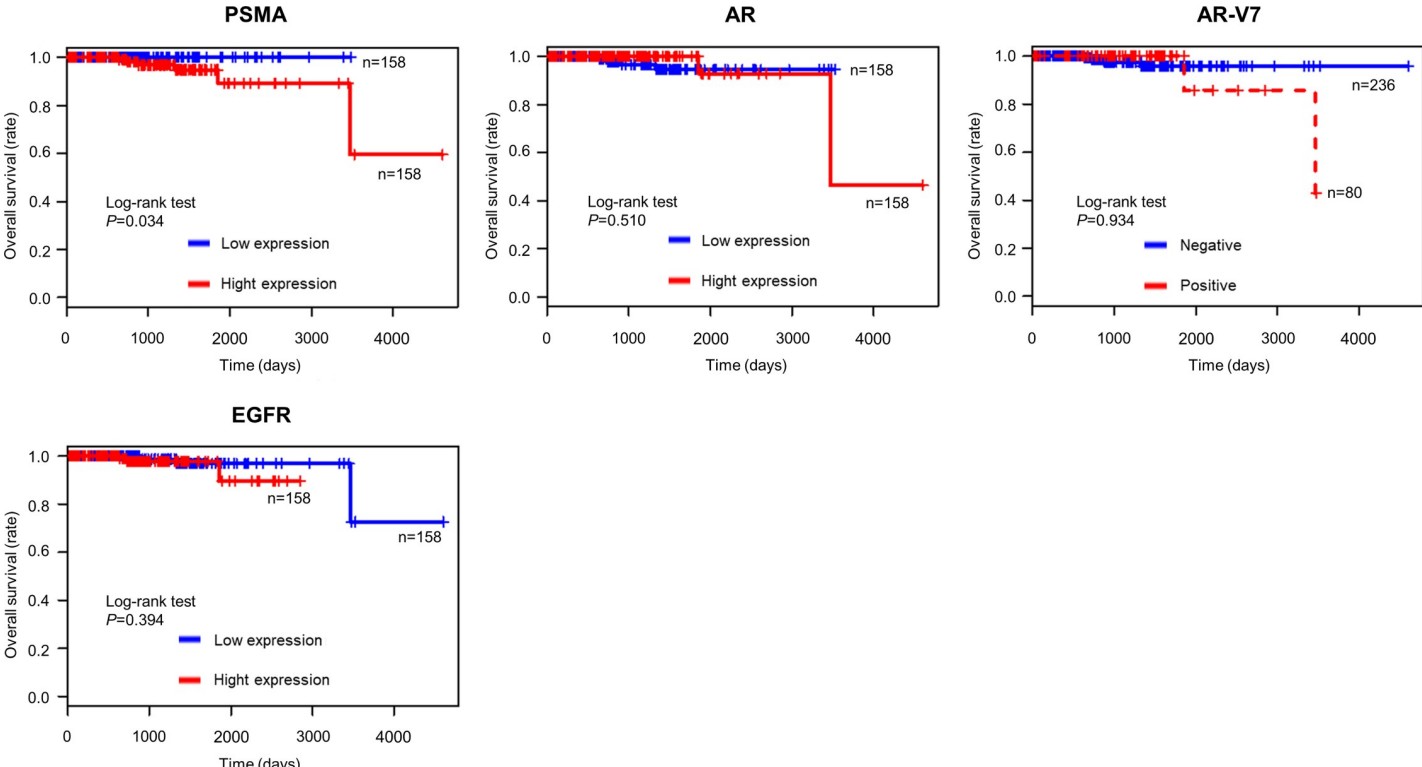

**Fig 6. The association between overall survival and mRNA expressions in TCGA cohort.** The Kaplan-Meier plot of overall survival was plotted according to each mRNA expression of primary prostate cancer in TCGA cohort (N = 316). Patients with prostate cancer expressing higher PSMA showed significantly shorter overall survival (*P* = 0.034).

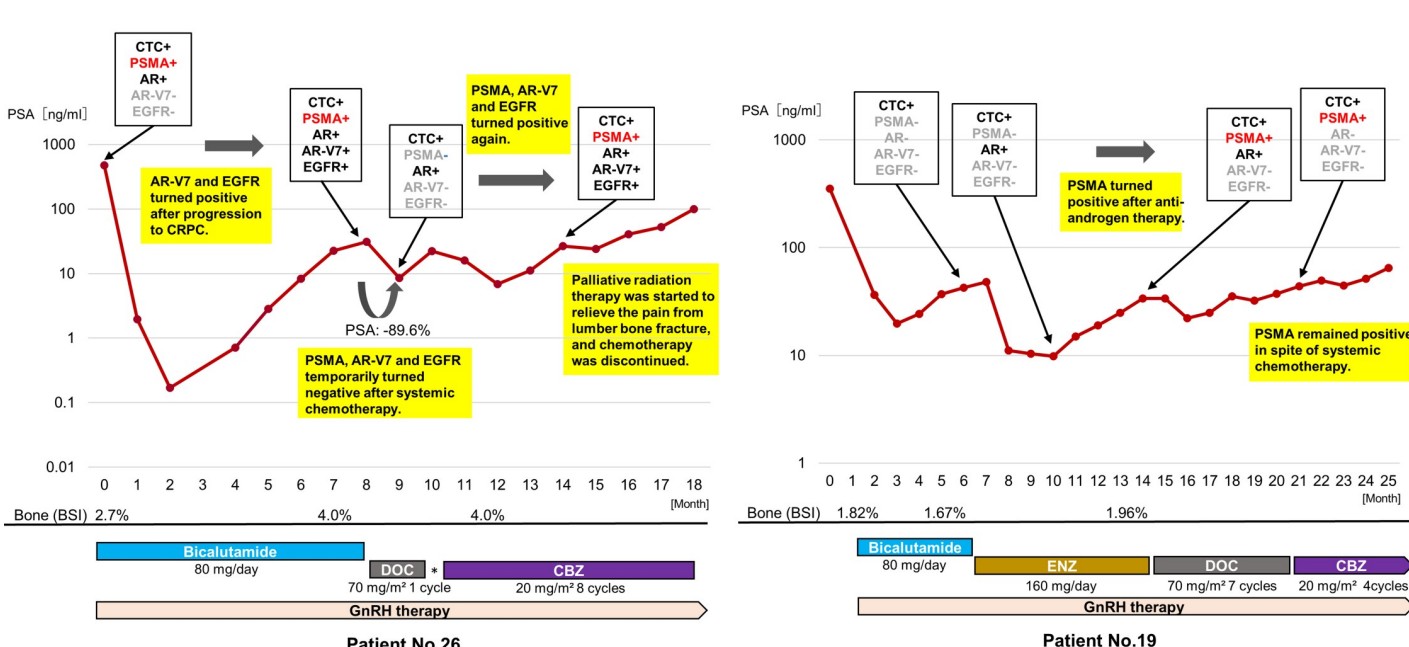

**Fig 7. Serial CTC monitoring of individual cases.** CTCs' gene expressions, PSA change, and treatment courses are designated in each chart. The *y*-axis is PSA value (ng/ml), and the *x*-axis is a time course. The percentage of change in PSA after treatment change is designated as actual numbers. **A.** Patient 26 was a case with positive PSMA expression that showed a rapid disease progression. Despite the early introduction of docetaxel, this patient moved on to palliative care within two years from diagnosis. **B.** This is a case that showed disease progression with a concurrent positive conversion of PSMA. Patient 19 who was initially negative for both PSMA and AR-V7 responded to enzalutamide. This patient showed PSA progression with a concurrent positive conversion of PSMA. In spite of systemic chemotherapies, the disease progressed and PSMA remained positive.

## Supporting information

**S1 Fig.**
(TIF)

## Acknowledgments

We would like to thank Dr. Yuji Tanaka (RIKEN) and the staff at Kyodo-ken for technical assistance.

## Author Contributions

**Conceptualization:** Shigeo Horie.

**Data curation:** Qi Hou, Zen-u Hotta.

**Formal analysis:** Zen-u Hotta.

**Funding acquisition:** Shigeo Horie.

**Investigation:** Naoya Nagaya, Masayoshi Nagata, Yan Lu, Mayuko Kanayama.

**Resources:** Toshiyuki China, Kosuke Kitamura, Kazuhito Matsushita, Shuji Isotani, Satoru Muto.

**Supervision:** Yoshiro Sakamoto.

**Validation:** Yoshiro Sakamoto.

**Writing – original draft:** Naoya Nagaya, Zen-u Hotta.

**Writing – review & editing:** Toshiyuki China, Kosuke Kitamura, Kazuhito Matsushita, Shuji Isotani, Satoru Muto, Yoshiro Sakamoto, Shigeo Horie.

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
