## [Decision Letter · Decision Letter 0]

7 Oct 2019

PONE-D-19-22397

Prostate-specific membrane antigen in circulating tumor cells is a new poor prognostic marker for castration-resistant prostate cancer

PLOS ONE

Dear Prof Horie,

Thank you for submitting your manuscript to PLOS ONE. After careful consideration, we feel that it has merit but does not fully meet PLOS ONE’s publication criteria as it currently stands. Therefore, we invite you to submit a revised version of the manuscript that addresses the points raised during the review process.

We would appreciate receiving your revised manuscript by Nov 21 2019 11:59PM. To enhance the reproducibility of your results, we recommend that if applicable you deposit your laboratory protocols in protocols.io, where a protocol can be assigned its own identifier (DOI) such that it can be cited independently in the future. For instructions see: http://journals.plos.org/plosone/s/submission-guidelines#loc-laboratory-protocols

We look forward to receiving your revised manuscript.

Kind regards,

Isaac Yi Kim, MD, PhD

Academic Editor

PLOS ONE

Journal Requirements:

1. Thank you for including your competing interests statement; "MN and SH received research grants and honorarium from Astellas, Sanofi, AstraZeneca, Takeda, Janssen. "

We note that you received funding from a commercial source: [Name of Company]

Reviewers' comments:

Reviewer's Responses to Questions

**Comments to the Author**

1. Is the manuscript technically sound, and do the data support the conclusions?

Reviewer #1: Yes

Reviewer #2: Yes

2. Has the statistical analysis been performed appropriately and rigorously? 

Reviewer #1: Yes

Reviewer #2: Yes

3. Have the authors made all data underlying the findings in their manuscript fully available?

Reviewer #1: Yes

Reviewer #2: Yes

4. Is the manuscript presented in an intelligible fashion and written in standard English?

Reviewer #1: Yes

Reviewer #2: Yes

5. Review Comments to the Author

Reviewer #1: his manuscript is studies on PSMA expression in CTCs as a poor prognostic marker for CRPC. Overall, this paper is well written. However, I have only several minor comments.

Minor comments:

1) Please indicate the number of patients instead of the number of samples to the abstract part.

Please indicate the number of patients instead of the number of CTC expression samples to the abstract.

Line 9 of page 2, 127/203 samples → ?/79 patients

2) Line 9 of page 2, 62% → 63% ?

3) It would be better to add a general description of CTCs to the introduction part.

4) Page 6, Please present PCR conditions and primer sequences for genes (PSMA, PSA, AR or EGFR) other than AR-V7.

If possible, I would like to see representative electrophoresis pictures of PCR products.

5) Line 12 of page 9, Thirty-six → Thirty-nine ?

6) Line 14 of page 14, Three of the five → Three of the six ?

7) When some abbreviations are shown in the text and figure at, please clarify the whole term.

For example: EP, LH-RH, ALP, LDH, ENZ, ABI, DOC, CBZ, PE

Reviewer #2: In this manuscript, Nagaya et al. first report a clinical significance of PSMA expression statuses of circulating tumor cells as a novel marker to predict the prognosis in castration resistant prostate cancer patients. They investigate the sensitivity of PSMA (62%) in 203 CTCs and revealed an inverse correlation with PSA expression and clearly showed that PSMA-positive patients with poor progression-free survival. These results suggest potential of clinical application to detect PSMA in CTCs. Although there are some remaining points before acceptance for publication in PLoS One, I am recommending that the editor request several minor improvements of the manuscript that I am outlining below before acceptance.

1. The description of treatment-switch (TS) cohorts is not adequate. What “Switch” means in this study? Please put a Supplementary Figure to specify and illustrate their time course.

2. Using the method of AdnaTest in this work, is it possible to detect spliced variants of PSMA?

3. I could not access the cBioPortal for Cancer Genomics from the URL described in the manuscript. Could you make sure whether this URL works properly?

4. In Figure 2C, CTC detection is a good predictor of PSA progression-free survival but not of overall survival. Please mention and discuss this discrepancy.

5. Color Keys of CTC Gene Expression in Figure 4 is so complicated. Please make them simpler such as to change to the gradation pattern.

6. In Figure 6, the authors divided all patients equally into two groups by expression level. Is it statistically valid considering the distribution of expression levels?

6. PLOS authors have the option to publish the peer review history of their article (what does this mean?). If published, this will include your full peer review and any attached files.

Reviewer #1: No

Reviewer #2: No

---

## [Author Response · Author response to Decision Letter 0]

21 Nov 2019

Dear Prof. Isaac Yi Kim,

We would like to thank you for inviting us to submit a revised draft of our manuscript entitled, “Prostate-specific membrane antigen in circulating tumor cells is a new poor prognostic marker for castration-resistant prostate cancer” to PLOS ONE. We also appreciate the time and effort you and each of the reviewers have dedicated to providing insightful feedback on ways to strengthen our paper. Thus, it is with great pleasure that we resubmit our article for further consideration. We have incorporated changes that reflect the detailed suggestions you have graciously provided. We also hope that our edits and the responses we provide below satisfactorily address all the issues and concerns you and the reviewers have noted.

To facilitate your review of our revisions, the following is a point-by-point response to the questions and comments delivered in your letter.

Responses to Reviewer 1’s Comments:

Comments to the Author 

his manuscript is studies on PSMA expression in CTCs as a poor prognostic marker for CRPC. Overall, this paper is well written. However, I have only several minor comments.

Response: 

We thank for carefully reading our manuscript and for giving useful comments. We have revised our manuscript on the basis of your comments.

#1

Please indicate the number of patients instead of the number of samples to the abstract part.

Please indicate the number of patients instead of the number of CTC expression samples to the abstract.

Line 9 of page 2, 127/203 samples → ?/79 patients

Response:

Thank you for your suggestion. We have revised the text to “CTCs were detected in 55/79 patients and median serum PSA in CTC-positive patients was 67.0 ng/ml.” at line 9-10 on page 2.

#2

Line 9 of page 2, 62% → 63% ?

Response: 

Thank you for pointing it out. I corrected that sentence.

#3

It would be better to add a general description of CTCs to the introduction part.

Response: 

Thank you for your valuable opinion. I have added the following sentence, “Since tissue biopsy is invasive and cannot be performed for all tumors, liquid biopsy is drawing attention as a new testing option to overcome these shortcomings. Liquid biopsies are the blood-based analyses of non-solid biological tissues such as exosomes, cell-free DNA and circulating tumor cells (CTCs).　CTCs are characterized as cancer cells that intravasate into the circulatory system from primary locations.” in introduction section at line 3-7 on page 3.

#4

Page 6, Please present PCR conditions and primer sequences for genes (PSMA, PSA, AR or EGFR) other than AR-V7.

If possible, I would like to see representative electrophoresis pictures of PCR products.

Response: 

Thank you for your suggestion. I have added the following sentence, “The AdnaTest PrimerMix ProstateDetect was used for amplification of PSA, PSMA, and EGFR (PCR condition for PSA, PSMA, and EGFR: 95 ℃ for 15 min, 42 cycles of 94 ℃ for 30 sec, 61 ℃ for 30 sec, 72 ℃ for 30 sec, followed by 10 min of extension). The AdnaTest PrimerMix AR-Detect was used for amplification of AR (PCR condition for AR: 95 ℃ for 15 min, 35 cycles of 94 ℃ for 30 sec, 60 ℃ for 30 sec, 72 ℃ for 60 sec, followed by 10 min of extension).” in methods section at line 3-9 on page 6. The representative electrophoresis images are listed below.

(The attached file)

#5

Line 12 of page 9, Thirty-six → Thirty-nine ?

Response: 

Thank you for pointing it out. I corrected that sentence.

#6

Line 14 of page 14, Three of the five → Three of the six ?

Response: 

Thank you for pointing it out. I corrected that sentence.

#7

When some abbreviations are shown in the text and figure at, please clarify the whole term.

For example: EP, LH-RH, ALP, LDH, ENZ, ABI, DOC, CBZ, PE

Response: 

Thank you for pointing it out. I added original names of each abbreviation.

Responses to Reviewer 2’s Comments:

Comments to the Author 

In this manuscript, Nagaya et al. first report a clinical significance of PSMA expression statuses of circulating tumor cells as a novel marker to predict the prognosis in castration resistant prostate cancer patients. They investigate the sensitivity of PSMA (62%) in 203 CTCs and revealed an inverse correlation with PSA expression and clearly showed that PSMA-positive patients with poor progression-free survival. These results suggest potential of clinical application to detect PSMA in CTCs. Although there are some remaining points before acceptance for publication in PLoS One, I am recommending that the editor request several minor improvements of the manuscript that I am outlining below before acceptance.

Response: 

Thank you for your time and effort in reviewing the paper. I would like to revise and answer your comments.

#1

The description of treatment-switch (TS) cohorts is not adequate. What “Switch” means in this study? Please put a Supplementary Figure to specify and illustrate their time course.

Response: 

Thank you for your precious opinion. I have added the Supplementary Figure 1 at line 8 on page 4 to illustrate what “all samples analysis” and “treatment-switch (TS) cohorts analysis” mean in this study.

#2

Using the method of AdnaTest in this work, is it possible to detect spliced variants of PSMA?

Response: 

You have raised an important question. We think it would be interesting to identify PSMA splicing variants. Although primers for PSMA splice variants are not included in AdnaTest kit itself, given that primers for AR-V7 (splice variant of AR) are custom-ordered, it is possible to perform RT-PCR for gene of interest with any arbitrary primer sets. However, because cDNA synthesized with this procedure was rather scarce, we focused on PSMA, PSA, AR, EGFR and AR-V7 in this study. Meanwhile, to further dissect CTC expression profile, we are currently trying to perform RNA sequencing with mRNA extracted by Adnatest. This method allows for not only detection of PSMA splice variants but also comprehensive expression profiling of CTCs. Thus, it is our future assignment to investigate the important agenda raised by the reviewer. 

#3

I could not access the cBioPortal for Cancer Genomics from the URL described in the manuscript. Could you make sure whether this URL works properly?

Response: 

Thank you for pointing it out. I corrected that URL.

#4

In Figure 2C, CTC detection is a good predictor of PSA progression-free survival but not of overall survival. Please mention and discuss this discrepancy.

Response:

Thank you for your suggestion. I have added the following sentence, “Although previous reports have shown that CTC-positive patients have significantly shorter overall survival, the presence of CTC was not a good indicator of overall survival in our study. Other reports using AdnaTest define the presence of CTC as the expression of any of PSA, PSMA, EGFR, or AR. Meanwhile, in our study, prostate cancer CTCs were categorized into only two types: PSA + / PSMA + or PSA + / PSMA-. This means that our CTC stratification based on PSMA is more detailed than other inclusive definition of CTC positivity. Thus, a possible explanation for this discrepancy is that CTC stratification based on PSMA better predicts overall survival than CTC positivity alone in a relatively small cohort like this study.

at line 2-10 on page 21.

#5

Color Keys of CTC Gene Expression in Figure 4 is so complicated. Please make them simpler such as to change to the gradation pattern.

Response: 

Thank you for your suggestion. I made them simpler, using 2 colors.

#6

In Figure 6, the authors divided all patients equally into two groups by expression level. Is it statistically valid considering the distribution of expression levels?

Response: 

Thank you for your valuable opinion. Median was used to generate a cutoff point to divide mRNA expressions into positive and negative. This method is often used when analyzing molecules with no established cutoff points for their expression to predict prostate cancer progression.

Again, we would like to thank you all for giving us the opportunity to further improve our manuscript with your informative comments and queries. We have worked hard to incorporate your feedbacks and hope that these revisions persuade you to accept our submission.

Sincerely,

Shigeo Horie

Professor and Chairman, Department of Urology

Juntendo University, Graduate School of Medicine

2-1-1 Hongo Bunkyo-ku, Tokyo, Japan 1138431

Tel and Fax: +81-3-5802-1227

E-mail: shorie@juntendo.ac.jp

---

## [Editor Report · Decision Letter 1]

22 Nov 2019

Prostate-specific membrane antigen in circulating tumor cells is a new poor prognostic marker for castration-resistant prostate cancer

PONE-D-19-22397R1

Dear Dr. Horie,

We are pleased to inform you that your manuscript has been judged scientifically suitable for publication and will be formally accepted for publication once it complies with all outstanding technical requirements.

With kind regards,

Isaac Yi Kim, MD, PhD

Academic Editor

PLOS ONE
---

## [Editor Report · Acceptance letter]

27 Nov 2019

PONE-D-19-22397R1 

Prostate-specific membrane antigen in circulating tumor cells is a new poor prognostic marker for castration-resistant prostate cancer 

Dear Dr. Horie:

I am pleased to inform you that your manuscript has been deemed suitable for publication in PLOS ONE. Congratulations! Your manuscript is now with our production department. 

With kind regards,

on behalf of

Dr. Isaac Yi Kim 

Academic Editor

PLOS ONE